# Interview Evaluation: A Novel Approach for Automatic Evaluation of Conversational Question Answering Models

**Xibo Li[1], Bowei Zou[2], Yifan Fan[1], Yanling Li [1], Ai Ti Aw[2], Yu Hong[1]***

[1]School of Computer Science and Technology, Soochow University, Suzhou, China
[2]Institute for Infocomm Research, A*STAR, Singapore
{cipolee20000210, yifanfannlp, li4861988, tianxianer}@gmail.com
{zou_bowei, aaiti}@i2r.a-star.edu.sg

## Abstract

Conversational Question Answering (CQA) aims to provide natural language answers to users in information-seeking dialogues. Existing CQA benchmarks often evaluate models using pre-collected human-human conversations. However, replacing the model-predicted dialogue history with ground truth compromises the naturalness and sustainability of CQA evaluation. While previous studies proposed using predicted history and rewriting techniques to address unresolved coreferences and incoherencies, this approach renders the question self-contained from the conversation. In this paper, we propose a novel automatic evaluation approach, interview evaluation. Specifically, ChatGPT acts as the interviewer (Q agent) with a set of carefully designed prompts, and the CQA model under test serves as the interviewee (A agent). During the interview evaluation, questions are dynamically generated by the Q agent to guide the A agent in predicting the correct answer through an interactive process. We evaluated four different models on QuAC and two models on CoQA in our experiments. The experiment results demonstrate that our interview evaluation has advantages over previous CQA evaluation approaches, particularly in terms of naturalness and coherence. The source code is made publicly available.[1]

## 1 Introduction

Conversational Question Answering (CQA) expects machines to answer questions in conversation given the evidence passage. Existing CQA datasets, such as QuAC (Choi et al., 2018), CoQA (Reddy et al., 2019), and DoQA (Campos et al., 2020), are derived from human-human conversations, where the questioner poses questions to gain knowledge on a specific topic, and the answerer responds based on the evidence passage and the conversation history. While significant progress

has been made in modeling CQA systems in recent years, the evaluation of CQA tasks has received less attention and remains flawed.

The prior CQA evaluation asks model questions in turn based on the held-out human-human conversations due to the high cost of involving humans during the evaluation process. However, this automatic evaluation method utilizes golden answers from the conversation history, disregarding the actual predictions made by the model. It has been noted by Mandya et al. (2020) that it deviates from real-world scenarios. They suggest using the models' own predictions for automatic evaluation. On the other hand, Li et al. (2022a) highlights that using predicted history can introduce issues such as unresolved coreference and incoherence and detect invalid questions by employing a coreference resolution model (Lee et al., 2018) and subsequently rewrite these questions by substituting incorrect mentions with correct ones.

Although the evaluation based on prediction combined with a rewriting mechanism mitigates the gap with real-world scenarios and has been proven to align better with human evaluation, the substitution in rewriting makes the conversational question self-contained, which destroys the coherence of the conversation. Furthermore, clarification is a significant feature in dialogues (Yatskar, 2019), where questioners often reformulate their questions if they are not satisfied with the previous answers received. Building upon this observation, we propose a novel evaluation method called interview evaluation, which takes into account the dynamic nature of conversations.

During an interview, interviewers often rely on a set of pre-prepared questions to assess interviewees. However, they also have the flexibility to generate new questions based on the interviewee's answers and provide prompts to guide them toward the correct response. Similarly, our interview evaluation for CQA simulates this human interview process.

---

*Corresponding author.

[1]https://github.com/cipolee/ interview_evaluation.

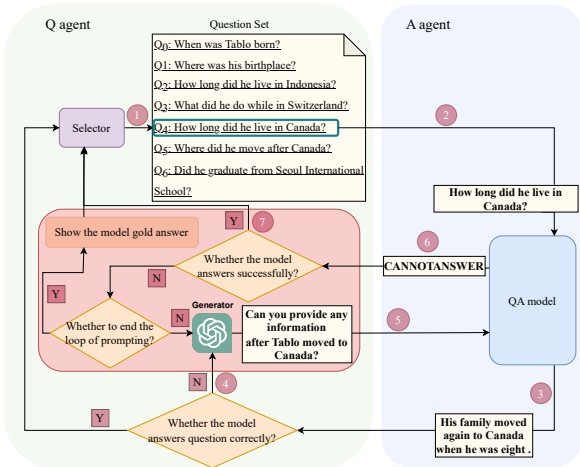

Figure 1: Architecture of interview evaluation.

It involves a Q agent that acts as the interviewer and an A agent representing the CQA model as the interviewee. This setup allows us to dynamically generate questions and prompts, mimicking the interactive nature of real interviews. Specifically, the Q agent consists of a selector and a generator. The selector sequentially chooses the next question from the question set, assuming the previous question was answered correctly. Otherwise, the generator iteratively generates new questions to prompt the CQA model until a question is marked as successful or unsuccessful.

We summarize our contributions as follows:

1) We propose a novel approach, interview evaluation, to assess CQA models by simulating a realistic and interactive interview scenario. This evaluation framework incorporates a question-generation agent (Q agent) that dynamically generates questions based on the CQA model's predictions.

2) We propose three metrics specifically tailored for interview evaluation, allowing for a multifaceted analysis of the models' performance from different perspectives.

3) We first introduce the ability of large language models in conducting CQA evaluation without the need for costly data collection of new datasets.

## 2 Preliminary

### 2.1 Existing Evaluation for CQA

The current typical evaluation process for CQA can be described as follows: In the $i$-th dialogue turn, a CQA model $\mathcal{M}$ is expected to answer a given question $Q_i$, based on the provided reference passage $\mathcal{P}$ and the conversation history $H_{i-1} = (Q_1, A_1, \cdots, Q_{i-1}, A_{i-1})$. It is worth noting that $Q_i$ is typically generated by a human, taking into account both $\mathcal{P}$ and $H_{i-1}$.

An ideal evaluation method for CQA systems would involve human participation, where humans can reflect on the conversation history and initiate a new round of questioning and answering. However, such way is accurate but also expensive. As a result, existing benchmarks rely on collections of human-human conversations for automatic evaluation, using golden history as a reference. Nevertheless, Mandya et al. (2020) argue that such evaluations do not truly reflect the model's performance as they overlook the models' own predictions. However, when incorrect predictions occur in conversational history, it leads to unresolved coreferences and incoherence in the current questions. To tackle this issue, Li et al. (2022a) propose two approaches: "rewriting" or "replacing" invalid questions. In "rewriting", entity names in invalid questions are substituted with correct entities, while in "replacing", the invalid question is replaced with a context-independent question sourced from CANARD (El-gohary et al., 2019). Figure 2 shows a case of "rewriting" where the question *When did they release the album?* is rewritten as *When did they release In This Light?* when the model incorrectly answers the question *What was In This Light?*.

However, the process of rewriting and replacing questions leads to paraphrasing, resulting in the loss of conversational question characteristics. As a result, these approaches still fall short of accurately reflecting the models' performance in real-world scenarios. This raises a crucial question: **How can we preserve the dialogical properties, such as reference and omission, in evaluation schemes?** This question emphasizes the need to maintain the conversational nature of the questions during the evaluation process.

### 2.2 CQA Models for Verification

To assess the effectiveness of interview evaluation and conduct further analysis, we adopt the approach proposed by (Li et al., 2022a) and select four CQA models that exhibit variations in both model structures and training methods. More details of the CQA models are in Appendix B. For comparison, we initially train these models on QuAC and CoQA, following the original implementations. However, the answer format of CoQA is free-form and may not necessarily be present in the given passage, which makes it impractical to

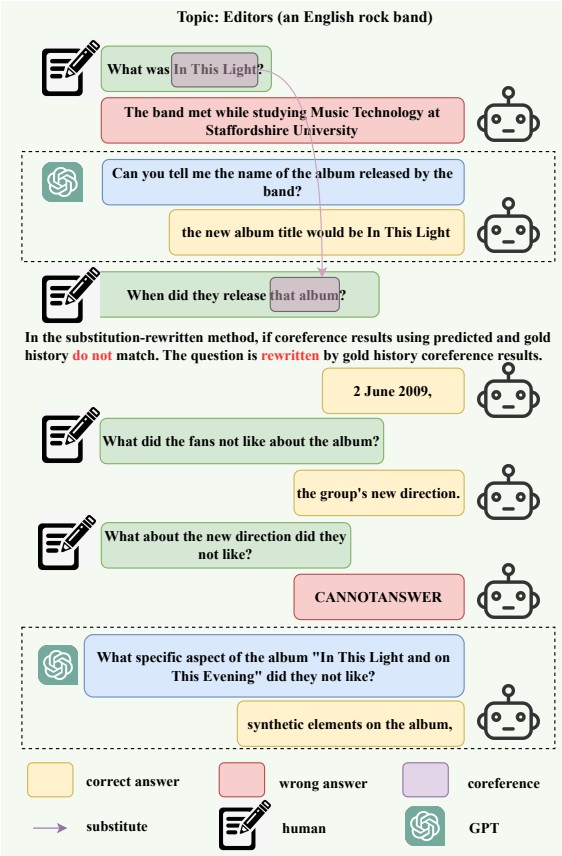

Figure 2: An example illustrating the generated-history interview evaluation. The green-colored questions are from the CQA dataset, while the blue-colored questions are generated by the Q agent. Based on the CQA model's predictions, the Q agent employs different strategies to determine the subsequent question.

apply the posHAE component of HAM. In addition, CoQA lacks a rewriting dataset like QuAC, which hinders the implementation of the question rewriting model utilized in ExCorD. As a result, our evaluation of CoQA is limited to using BERT and GraphFlow.

We adopt different evaluation settings following Li et al. (2022a): *Auto-golden* for evaluation using golden history, *Auto-Pred* for evaluation based on model predictions, *Auto-Rewrite* for evaluation involving question rewriting, and *Auto-Replace* for evaluation utilizing replacing. Subsequently, we assess the performance of CQA models in *Auto-golden* and *Auto-Pred*. We also incorporate the human performance results reported by Li et al. (2022a), obtained through interactively questioning and checking model predictions at Amazon Mechanical Turk. Furthermore, we analyze the correlation between our proposed interview evaluation and human evaluation.

## 3 Interview Evaluation

### 3.1 Interview Process Imitation

Interviews have the nature of conversational question answering, where interviewers typically select questions from a pre-prepared set to assess the interviewee. In addition, interviewers often generate new questions based on the interviewee's responses and provide prompts to guide them toward the correct answers. The introduction of interview datasets in programming, such as APPS (Hendrycks et al., 2021), has contributed to advancements in the field. Expanding on this, Zhang et al. (2023) propose the utilization of key prompts, such as compile or runtime error messages derived from test results, to enhance debugging and problem-solving in code generation models. To simulate realistic human behavior, Park et al. (2023) introduce agents that rely on generative models. These agents' capabilities, including memory and planning, are evaluated through interviews covering various question categories.

Inspired by Zhang et al. (2023); Park et al. (2023), we propose an evaluation method that simulates an interview process for assessing CQA models. During an interview, the interviewer evaluates the interviewee's knowledge by assessing the correctness of their answers. In addition, the interviewer generates new questions based on the interviewee's responses and provides prompts to guide them toward the correct answers when they respond incorrectly or answer with "unknown". In our proposed interview evaluation for CQA, we take the Q agent as the interviewer and the A agent as the CQA model being tested. The Q agent evaluates the abilities of the A agent from two perspectives: (1) the number of prompts required for the A agent to answer a question correctly, which reflects their comprehension of questions and utilization of knowledge (i.e., evidence passage), and (2) the ratio of the A agent providing correct answers under prompting, indicating their understanding of the conversational context. This ability is further demonstrated by the ratio of transitioning from responding "unknown" to answering correctly. A higher-performing A agent is characterized by the ability to answer questions correctly with fewer prompts and a lower ratio of not being able to answer questions correctly under prompts.

The framework for interview evaluation is illustrated in Figure 1. The Q agent consists of three modules: a question set, a selector, and a generator.

The question set encompasses a series of sequential questions derived from previously gathered human-human conversations. The selector functions as a state execution mechanism, determining the next question in the set based on specific triggers within the state. The generator generates new questions based on the model's predictions.

For each conversation, the selector initially selects a question from the beginning of the pre-collected human question set. If the model (A agent) answers correctly, the selector proceeds to the next question. However, if the model answers incorrectly, the generator generates a new question. Notably, in cases where the model answers incorrectly, the generator persists in generating questions as prompts until two exit states are encountered: *success* and *failure*. The *success* state occurs when the model generates a correct answer with a higher F1 score between the golden answer than a configurable threshold. The *failure* state is further divided into two scenarios: 1) when the model refuses to answer (e.g., responding with CANNOTANSWER in QuAC or unknown in CoQA), and 2) when the number of newly generated questions reaches the upper limit, yet the model still answers incorrectly.

### 3.2 Methodology and Metric

The centerpiece of interview evaluation is the Q agent, which generates questions to elicit model predictions and guide the model to provide correct answers. Rather than disregarding the model predictions and asking the question directly in the subsequent turn, the generator $\mathcal{G}$ within the Q agent generates multiple questions to steer the model towards providing a correct answer corresponding to the original human question. Therefore, unlike previous evaluations, the conversation in the interview evaluation is dynamically generated based on the model predictions. Let $Q_i^*$, $A_i^*$, and $C_i^*$ represent the human question, human answer, and a golden conversation turn, respectively.

$$C_i^* = (Q_i^*, A_i^*) \qquad (1)$$

Let $Q_i^j$ represent the $j$-th new question generated by $\mathcal{G}$ for $Q_i^*$, and let $A_i^j$ denote the model prediction for $Q_i^j$. Specifically, the model prediction for $Q_i^*$ is denoted as $A_i^0$. The generated conversation cell for $Q_i^*$ is represented as:

$$C_i = \left(Q_i^*, A_i^0, Q_i^1, A_i^1, \cdots, Q_i^n, A_i^n\right), \qquad (2)$$

where $n$ represents the maximum number of generator prompts. Similarly, we use $C_i^j$ to denote the conversation cell up to $Q_i^j$. The golden history is denoted as:

$$H_i^* = (C_0^*, C_1^*, \cdots, C_i^*) \qquad (3)$$

Similar to $H_i^*$, the history generated by model is represented as:

$$H_i = (C_0, C_1, \cdots, C_i) \qquad (4)$$

For the original question $Q_i^*$, if the model predicts an incorrect answer, new questions are generated to prompt the model. In the $j$-th turn of prompting, the generator $\mathcal{G}$ generates a new question $Q_i^j$ based on the generated history $H_{i-1}$, the conversation cell $C_i^{j-1}$ for $Q_i^*$ and the golden answer $A_i^*$:

$$Q_i^j = \mathcal{G}\left(H_{i-1}, C_i^{j-1}, A_i^*\right) \qquad (5)$$

Correspondingly, the model predicts an answer $A_i^j$, given the generated history $H_{i-1}$, the conversation cell $C_i^{j-1}$ and the new question $Q_i^j$. We name it as **generated-history interview evaluation**.

$$A_i^j = \mathcal{M}\left(\mathcal{P}, H_{i-1}, C_i^{j-1}, Q_i^j\right) \qquad (6)$$

For comparison, we also explore another way that utilizes the golden history for model evaluation, without the questions generated by the generator $\mathcal{G}$ and their corresponding model answers. We name it as **golden-history interview evaluation**. In contrast to the intricate and conflicting nature of the generated history, the golden history represents a simple and ideal scenario of the generated history:

$$A_i^j = \mathcal{M}\left(\mathcal{P}, H_{i-1}^*, C_i^{j-1}, Q_i^j\right) \qquad (7)$$

The interview evaluation compares models in three aspects. Both the golden-history interview evaluation and the generated-history interview evaluation utilize three metrics to evaluate CQA models. The first metric is the average number of prompt questions required for each round of answering until the model answers correctly or reaches the maximum number of questions, referred to as **Questions Per Round (QPR)**. Denoting the number of questions marked in the success state as $N_s$ and the number of questions generated by $\mathcal{G}$ as $N_g$, QPR is computed as follows:

$$QPR = \frac{N_s + N_g}{N_s} \qquad (8)$$

The second metric is the **Persistent Failure Rate (PFR)**, which measures the failure rate of

CQA models even after receiving multiple prompts from the Q agent. Denoting the number of questions marked in the failure state as $N_f$, PFR is computed as follows:

$$PFR = \frac{N_f}{N_s + N_f} \qquad (9)$$

The third metric is the **Answer Conversion Rate (ACR)**, which measures the rate at which the QA model transitions from incorrectly refusing to answer (e.g., incorrectly answering with unknown or CANNOTANSWER, indicating non-unanswerable) to correctly answering. Denoting the number of non-unanswerable questions that the model predicts incorrectly as $N_{ui}$, and the number of times the model correctly answers an initially non-unanswerable question under prompting as $N_{uc}$, ACR is computed as follows:

$$ACR = \frac{N_{uc}}{N_{ui}} \qquad (10)$$

Compared with solely using ChatGPT to score CQA model predictions, the evaluation with the three metrics we propose for the predictions is more reliable and fair.

## 4   Experimentation

### 4.1   Settings

**Dataset.** We conduct experiments on QuAC (Choi et al., 2018) and CoQA (Reddy et al., 2019). QuAC comprises 98k question-answer pairs, obtained from 13k conversations. It contains 20.2% unanswerable questions and the average length of answers is 14 tokens. CoQA comprises 127K question-answer pairs obtained from 8k conversations in seven domains. It contains only 1.3% unanswerable questions and contains more factoid questions, which results in an average answer length of 2.7 tokens. Due to the average dialogue length (number of dialogue turns) of CoQA being twice that of QuAC, we curate a test set consisting of 400 QuAC dialogues and 200 CoQA dialogues, comprising a total of 2.8k and 2.5k question-answer pairs, respectively.

**Agent Settings.** When the A agent provides the correct answer, the Q agent sequentially selects the next question from the pre-collected conversational question set to ask the A agent, otherwise the Q agent will generate new questions to guide the A agent to answer correctly, as illustrated in Appendix A. The maximum number of prompts in Q agent

is set to 3 and the maximum history window of A agent (CQA models) is set to 2. We implement interview evaluation using both golden history and generated history, with further details provided in Subsection 3.2.

**Prompt Engineering for Question Generation.** We conduct prompt tuning and compare the effect of prompts by manually checking the fluency, plausibility and relevance of generated questions. The final template consists of three components: (a) *task definition*, which introduces the interview task to ChatGPT and assigns the roles of the interviewer and interviewee; (b) *background*, which is a fill-in-the-blank text requiring the inclusion of the conversation history, current question, prediction and golden answer; (c) *instruction*, which guides the context generated by the language models. The template of prompts for generating new questions is as follows:

> Forget the instruction you have previously received. The following is an interview assessment where you play the role of the interviewer and the other person plays the role of the interviewee. The interviewee needs to incorporate conversation history to answer the current question. For questions that the interviewee cannot answer correctly, the interviewer needs to ask new questions with hints to guide the model to the correct answer without revealing it. The conversation history is {history}. The current question is {cur_question} and the interviewee answers {prediction}, however, the golden answer is {golden_answer}. Then you ask a new question with hint:

Then, we fill in the necessary information into this template and utilize the ChatGPT API to generate new questions, as shown in Figure 3.

### 4.2   Main Results

We conduct a comparison between the interview evaluation and the automatic evaluation, namely *Auto-golden* and *Auto-Pred*. Furthermore, we analyze the alignment between our proposed metrics and the human evaluation in QuAC. The human evaluation process entails a conversation comprising 8-12 questions posed by an annotator who is unaware of the passage content.

**Consistency of Evaluation Metrics.** The performance of the evaluated models on QuAC and

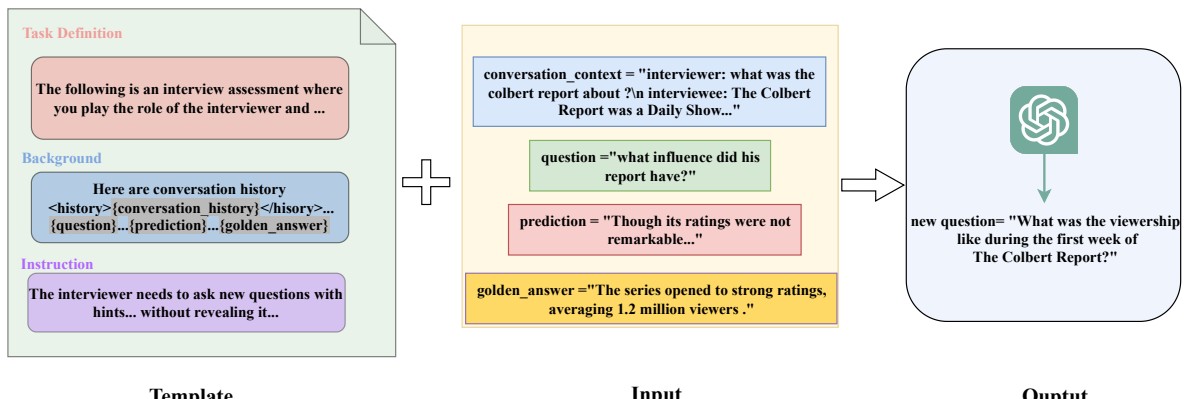

**Template**          **Input**          **Ouptut**

Figure 3: Process of prompt engineering for question generation using the ChatGPT API.

| | QuAC | | | | CoQA | |
|---|---|---|---|---|---|---|
| | BERT | GraphFlow | HAM | ExCorD | BERT | GraphFlow |
| Auto-golden (F1 %) | 61.5 | 65.1 | 64.4 | 66.9 | 78.1 | 76.1 |
| Auto-Pred (F1 %) | 53.0 | 50.7 | 56.2 | 60.3 | 70.1 | 62.8 |
| Interview Evaluation♣ (QPR) | 1.36 | 1.41 | 1.43 | 1.28 | 1.19 | 1.18 |
| - (PFR %) | 19.7 | 21.6 | 19.3 | 16.0 | 14.0 | 21.8 |
| - (ACR %) | 59.3 | 54.8 | 63.8 | 57.9 | 23.5 | 0 |
| Interview Evaluation◇ (QPR) | 1.31 | 1.37 | 1.28 | 1.24 | 1.16 | 1.17 |
| - (PFR %) | 18.3 | 20.6 | 15.4 | 15.6 | 12.8 | 21.8 |
| - (ACR %) | 54.9 | 54.5 | 64.6 | 56.1 | 29.7 | 0 |
| Human Evaluation (Accuracy %) (Li et al., 2022a) | 82.6 | 81.0 | 87.8 | 87.9 | - | - |

Table 1: Performance of evaluated CQA models on QUAC and CoQA. Note: ♣ indicates interview evaluation based on generated history; ◇ indicates interview evaluation based on golden history. The accuracy of Li et al. (2022a)'s evaluation on CoQA is missing as they only report human evaluation in QuAC. Note that model performance is negatively correlated with the values of QPR and PFR.

CoQA is listed in Table 1, and the key observations can be summarized as follows: 1) In the golden-history interview evaluation, when considering the metrics of QPR and PFR, the performance ranking on QuAC is ExCorD ≈ HAM > BERT > Graph-Flow. However, in the generated-history interview evaluation, the performance ranking changes due to the significant performance degradation of HAM while the rankings of other models remain consistent. 2) In terms of the ACR metric, whether in the golden-history interview evaluation or the generated-history interview evaluation, the performance ranking on QuAC is HAM > ExCorD > BERT > GraphFlow. The ranking of BERT and GraphFlow on CoQA is also consistent.

We verify that our proposed interview evaluation can provide more comprehensive insight into the strengths and weaknesses of CQA models through an analysis of the performance on evaluation metrics. 1) GraphFlow performs poorly in predicting unanswerable questions due to it utilizing a difficultly-calibrated separate network for unan-

swerable question predictions, which causes a drop in overall performance; HAM and ExCorD enhance the ability to understand the question in a conversational context and yield superior performance; ExCorD adopts a question rewriting module that generates context-independent questions based on conversational history; HAM emphasizes the mentions in questions and conversation history by special embeddings (turn marker and PosHAE) and uses a history attention mechanism to select the most relevant part from the history. 2) We notice that in the interview evaluation, the ACR improvement of HAM is the most obvious, which benefits from the unique history attention mechanism and special embeddings that make it easy for the model to correct the answer from unanswerable to answerable, and give a most suitable passage span as the answer. 3) However, HAM does not perform well in the other two metrics on generated-history interview evaluation. Our analysis is because the history generated by the Q agent and the A agent introduces a lot of conflict-laden history content,

| | B | G | H | E |
|---|---|---|---|---|
| Auto-golden (F1 %) | 60.3 | 63.7 | 64.2 | 65.1 |
| Auto-Pred (F1 %) | 50.1 | 49.1 | 53.9 | 57.7 |
| Interview Evaluation♣ (QPR) | 1.42 | 1.48 | 1.52 | 1.33 |
| - (PFR %) | 16.1 | 15.6 | 14.7 | 13.2 |

Table 2: Performance for answerable questions on QuAC. B: BERT; G: GraphFlow; H: HAM; E: ExCorD. ♣ indicates the evaluation based on generated history.

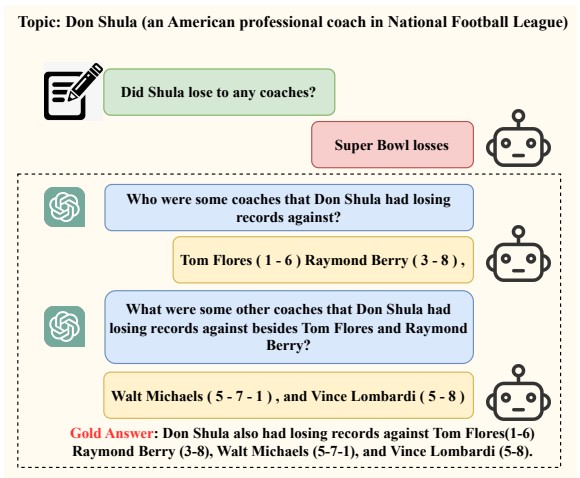

Figure 4: A bad case for interview evaluation.

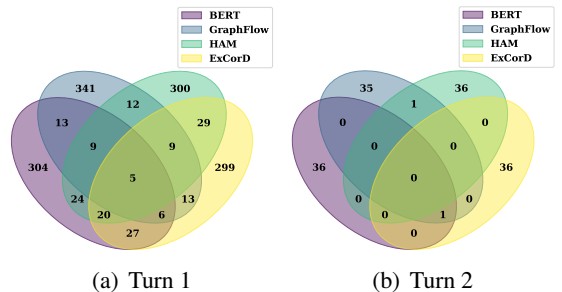

(a) Turn 1    (b) Turn 2

Figure 5: Overlap degree of new questions for the four CQA models. The numbers within the Venn diagram indicate the count of identical questions generated for the corresponding model by the Q agent.

which limits both the historical attention mechanism and special embeddings and reduces the ability to correct answers from a wrong passage span to a correct passage span.

**Results of Answerable CQA.** Considering the negligible impact of the 1.3% unanswerable questions in QuAC, we only show the results for answerable questions in Table 2. As the ACR metric remains the same for both answerable and overall questions, we refrain from redundant ACR reporting and focus on QPR and PFR. Based on these metrics, the CQA models are ranked as: for QPR, ExCorD > BERT > GraphFlow > HAM; and for PFR, ExCorD > HAM > GraphFlow > BERT.

We observe a notable decrease in the performance of GraphFlow from *Auto-golden* to *Auto-Pred*. It is attributed to GraphFlow modeling incorrect predictions in the conversation history, which introduces noise and hampers the understanding of the current question. On the other hand, the performance of HAM shows only slight changes. This is because HAM incorporates a history attention mechanism that selects the most relevant part of the history, helping to mitigate the impact of incorrect predictions. However, in the interview evaluation, where the conversation history is significantly expanded, HAM's performance on the QPR metric is affected. Nevertheless, HAM remains competitive in terms of the PFR metric. Its focus on the prompt related to the question enables it to make correct predictions during subsequent rounds of prompts. Therefore, compared to existing CQA evaluation methods, our interview evaluation provides a better assessment of the CQA model's ability to continuously understand dialogue and answer questions, bringing it closer to real-world scenarios. However, it's important to note that existing evaluation methods still rely on a single round of evaluation due to the use of golden history.

### 4.3 Case Study

We present two cases to illustrate the advantages and disadvantages of interview evaluation. In the first case depicted in Figure 2, the Q agent prompts the CQA model with more specific questions such as *Can you tell me the name ...* or *What specific aspect ... did they not like?* when the model provides an incorrect answer or responds with CANNOTANSWER. In comparison to the *Auto-Rewrite* approach, interview evaluation leverages the Q agent to guide the model toward providing correct answers, thereby avoiding the invalidation of questions. As a result, interview evaluation not only preserves the coherence of the dialogue but also evaluates the model's ability to reconsider the conversation history under the provided prompts.

Figure 4 illustrates a limitation of the interview evaluation, as it may not fully capture all correct predictions made by the model. In this case, the model accurately predicts *Tom Flores (1-6) Raymond Berry (3-8)*, but it is incorrectly misjudged as an incorrect answer because its prediction has a low overlap with all candidate answers. Future studies should focus on optimizing the prediction

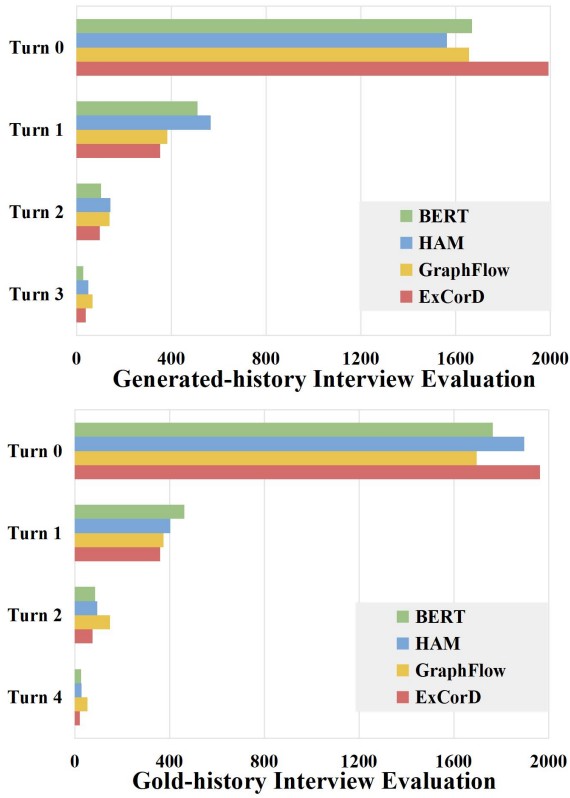

Figure 6: Number of prompts required to correctly answer questions on QuAC. "Turn 0" indicates the models' ability to accurately predict human questions, while "Turn $i$" indicates correctly predicting with the prompt of the $i$-th new question.

|           | QPR ♣  | PFR ♣ | QPR ◇ | PFR ◇ |
|-----------|--------|-------|-------|-------|
| Spearman  | -0.32  | 0.54  | -1.0  | -0.80 |
| Pearson   | -0.40  | 0.20  | -0.91 | -0.97 |

Table 3: Pearson and Spearman correlation coefficients between human evaluation and interview evaluation. ♣ and ◇ indicate the metrics for interview evaluation based on generated history and golden history, respectively.

assessing module and the question generator of the Q agent to account for cases where the model's answer is partially correct.

## 5 Analysis and Discussion

**Adaptability and Flexibility for Interview Evaluation.** One notable characteristic of interview evaluation is its ability to generate prediction-aware questions and can adapt to different evaluated CQA models. To investigate this aspect, we analyze the number of overlapping questions generated by ChatGPT across different models on QuAC. Specifically, we focused on prompts originating from the same human question and selectively collected conversation cells that contained these prompts for

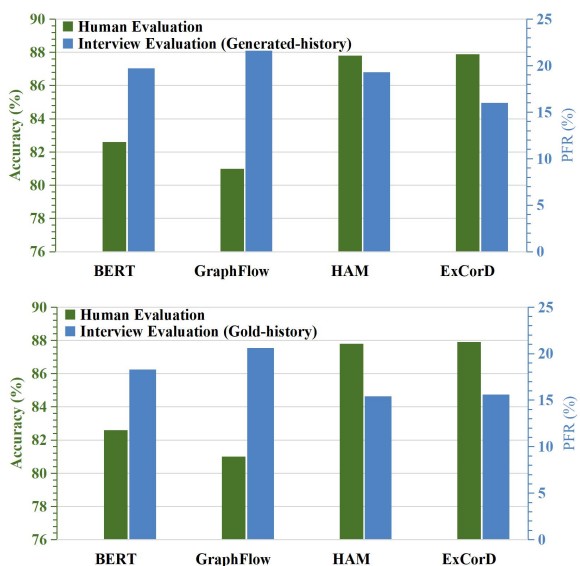

Figure 7: Accuracy of human evaluation vs PFR of interview evaluation.

analysis across all four models. Our analysis resulted in a dataset of 408 examples with a conversation cell length of at least 2, and 37 examples with a conversation cell length of at least 3. We considered questions to be overlapping if they were an exact match in terms of their text. To visualize the degree of overlap, we utilized a Venn diagram, which is presented in Figure 5. The diagram demonstrates that questions generated by ChatGPT for different models exhibit a low degree of overlap, and this degree decreases as the number of prompts increases. In conclusion, the generated questions used in the evaluation process are diverse and effectively reflect the predictions of the evaluated models. This diversity and alignment with model predictions represent a key advantage of the interview evaluation approach.

**Golden History vs. Generated History.** We conducted a comparison between two modes of interview evaluation: based on golden history and based on generated history. The number of prompts required to successfully answer a question for four CQA models on QuAC is reported in Figure 6. The results indicate that all models perform better on human-generated questions compared to the new questions generated by ChatGPT. It is evident that the questions requiring ChatGPT-generated prompts are more challenging for the models, as they have been answered incorrectly before. Notably, we observed that HAM performs worse than the other three models on Turn 0 in the generated-history interview evaluation. However, HAM per-

forms better when answering new questions generated by ChatGPT, benefiting from its history attention mechanism.

Furthermore, we analyze the PFR performance and compare it with the results of human evaluation, as depicted in Figure 7. The results demonstrate that the performance rankings of the golden-history interview evaluation and human evaluation align with each other. However, the performance rankings of the generated-history interview evaluation and human evaluation are not consistent. Compared to human evaluation, interview evaluation introduces a Q agent that is aware of model predictions and gives prompts. Therefore, the generated-history interview evaluation expands the dialogue history to make history full of conflict. Consequently, the generated interview evaluation scenarios are more intelligent and complex to better expose the limitations of the model such as HAM.

To gain a better understanding of the two evaluation modes, we analyze their agreement with human evaluation in Table 3. In terms of QPR, both the golden-history interview evaluation and human evaluation exhibit a strong agreement with a Spearman correlation coefficient of -1.0 and a Pearson correlation coefficient of -0.91. However, the correlation coefficients between the generated-history interview evaluation and human evaluation only range between -0.5 and 0. It indicates that the relationship between human evaluation and golden-history interview evaluations is strong, while the relationship with the generated-history interview evaluation is weak. The same conclusion applies to the PFR metric.

It is important to note that the high consistency between human evaluation and golden-history interview evaluations does not imply that the interview evaluation based on golden history is superior. The reason for this high consistency is that, both of them utilize the golden history, which means the CQA model does not have to account for its previous performance in the conversation. As we previously analyzed, the interview evaluation based on generated history provides a better assessment of the model's ability to continuously understand the conversation and the interactive ability, which are not adequately captured by existing CQA evaluation methods.

## 6 Conclusion

This paper presents the concept of interview evaluation as a novel approach for assessing CQA models. Our interview evaluation framework employs a question-generation agent (Q agent) to dynamically generate questions based on the model's predictions, facilitating a more realistic and interactive evaluation scenario. We introduce three metrics specifically designed for interview evaluation to assess models from different perspectives. Through our analysis, we compared two modes of interview evaluation: golden history and generated history. We also examined the performance of these modes with automatic evaluations and human evaluation.

However, it is important to acknowledge that the interview evaluation has its limitations. In some cases, it overlooks correct predictions when the model's output partially aligns with the correct answer. Besides, it is difficult to evaluate the diversity and creativity of answers provided by the generative model. Future research should focus on optimizing the answer-matching module and the question generator of the Q agent to handle such cases more effectively.

## Limitations

In interview evaluation, we determine a prediction as correct if the F1 score between the golden answer and the model's prediction exceeds a predefined threshold. However, relying solely on F1 scores can introduce bias towards longer answers and may not capture all correct answers, particularly shorter ones.

Additionally, our approach leverages LLMs to dynamically generate questions about model predictions. The introduction of LLMs also improves generalization since LLMs can be effectively adapted to downstream tasks by building few-shot task-specific prompts. However, LLMs suffer from the issue of overfitting on the training data. Therefore, the quality of the generated questions will also be affected, especially if the original questions contain the latest knowledge that is not used to train the LLMs. Moreover, the new questions generated by LLMs exhibit diversity for different incorrect predictions, and quantifying the difficulty of these questions becomes challenging.

## Acknowledgements

The research is supported by National Key R&D Program of China (2020YFB1313601), Na-

tional Science Foundation of China (62376182, 62076174) and Institute of Infocomm Research of A*STAR (CR-2021-001).

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

## A   Algorithm for Interview Evaluation

Algorithm 1 illustrates the process of interview evaluation.

## B   CQA Models for Test

To verify the effectiveness of our interview evaluation, we employ the following four CQA models and conduct them on QuAC and CoQA.

- **BERT**: To establish it, we construct a model by adding multiple linear layers on top of the encoder of BERT (Kenton and Toutanova, 2019). The input comprises the concatenation of two preceding rounds of historical question-answer pairs, along with the current question and the passage. The model is trained to predict the answer span.

---

**Algorithm 1** Interview Evaluation
1: Initialize Selector $\mathcal{S}$, Generator $\mathcal{G}$
2: Initialize a question set $\mathbb{Q}$ collected from the given human-human conversation
3: Set the maximum number of prompting to $N$
4: Initialize the index of question in $\mathbb{Q}$, $i = 0$
5: **while** $i < |\mathbb{Q}|$ **do**
6: $\quad$ $\mathcal{S}$ selects a question $q_i$ from $\mathbb{Q}$
7: $\quad$ **if** QA model answers correctly **then**
8: $\quad\quad$ Mark $q_i$ as a success state
9: $\quad$ **else**
10: $\quad\quad$ Initialize the number of prompting $j = 0$
11: $\quad\quad$ **while** no state marked in $q_i$ **do**
12: $\quad\quad\quad$ $\mathcal{G}$ generates a new question
13: $\quad\quad\quad$ **if** QA model answers correctly **then**
14: $\quad\quad\quad\quad$ Mark $q_i$ as a success state
15: $\quad\quad\quad$ **else if** $j > N$ or QA model predicts the question is unanswerable **then**
16: $\quad\quad\quad\quad$ Mark $q_i$ as a failure state
17: $\quad\quad\quad$ **end if**
18: $\quad\quad\quad$ $j = j + 1$
19: $\quad\quad$ **end while**
20: $\quad\quad$ **if** $q_i$ is marked as failure **then**
21: $\quad\quad\quad$ Show the model golden answer
22: $\quad\quad$ **end if**
23: $\quad$ **end if**
24: $\quad$ $i = i + 1$
25: **end while**

---

- **GraphFlow**: Chen et al. (2021) combine the reasoning layer on top of the encoding layer to capture rich semantic relations within the context. The reasoning layer adopts a recurrent graph neural network to process the output of the encoding layer.

- **HAM**: Qu et al. (2019) apply a history attention module to assign different weights to history turns and softly select the most relevant previous turns. Besides, The PosHAE is employed to map each token to a history answer embedding in the passage, which enhances history answer embedding by incorporating position feature of history turns.

- **ExCorD**: Drawing upon CANARD (Elgohary et al., 2019), Kim et al. (2021) introduce a question rewriting model to enhance CQA models' comprehension of conversational context.

## C  Related Work

### C.1  Conversational Question Answering

After several conversational question answering datasets have been proposed, such as QuAC (Choi et al., 2018) and CoQA (Reddy et al., 2019), the studies focusing on conversational question answering (Qu et al., 2019; Chen et al., 2021; Kim et al., 2021; Qian et al., 2022; Li et al., 2022b) have emerged. Recently, some studies have noticed that the current method of evaluating conversational QA models is flawed. Mandya et al. (2020); Siblini et al. (2021) point out that utilizing golden answer in history deviates from real-world scenarios and proposes to use predicted history. Li et al. (2022a) points out to rewrite these questions which suffer from unresolved coreference and incoherence. Different from prior works, in this paper, we propose a novel evaluation, namely interview evaluation, utilizing ChatGPT to generate dynamic and coherent dialogue. In the evaluation of CQA systems, prior studies have commonly employed the F1 score as the primary metric, similar to reading comprehension tasks such as SQuAD (Rajpurkar et al., 2016). The F1 score indicates the extent of lexical overlap between the model prediction and the golden answer.

### C.2  Prompting

Kojima et al. (2022); Radford et al. (2019) propose to utilize the prompts to direct models toward the given task or topic, thereby reducing the task of generating irrelevant or inaccurate answers. The prompts are divided into "hard" prompts made from interpretable words and tokens and "soft" prompts consisting of continuous feature vectors such as Prefix-Tuning (Li and Liang, 2021). (Shin et al., 2020) designs a hard template that acts as a fill-in-the-blank format where specific slots or placeholders are defined to represent missing information. This text-based prompt controls the generation of large models. Recent, The development of LLMs such as GPT3 (Brown et al., 2020) and PaLM (Chowdhery et al., 2022) facilitates the zero-shot instruction. Zero-shot instruction is highly effective due to the extensive data used to pre-train large-scale language models, enabling them to develop a comprehensive understanding of language and reasoning ability. Thus, we carefully craft a hard prompt to control ChatGPT with the instruction following capability to generate the text we require.