# OpenReview forum: "Interview Evaluation: A Novel Approach for Automatic Evaluation of Conversational Question Answering Models"
_EMNLP/2023/Conference — EMNLP 2023 Main_

### Official Review · Reviewer_deJG · 2023-08-04

**Typos Grammar Style And Presentation Improvements:** 1. Line 177
**Soundness:** 4

**Excitement:**

4: Strong: This paper deepens the understanding of some phenomenon or lowers the barriers to an existing research direction.

**Paper Topic And Main Contributions:**

The paper identifies and addresses a crucial yet unsolved problem of Conversational Question Answering (CQA) task. Current CQA benchmarks use pre-collected human-human conversation to evaluate models performance, however, this method ignores the actual predictions made by model. Also, other method use substitution and rewriting mechanism to mitigate the gap with real-world scenarios destroys the coherence of the conversation. The authors not only identify these flaws in existing evaluation methods but also propose the interview evaluation method which considers the dynamic and interactive nature of conversations, which is often neglected in other evaluation methodologies. There are three new metrics (Questions Per Round, Persistent Failure Rate, Answer Conversation Rate) specifically designed for interview evaluation, providing a multi-dimensional analysis of model performance. The experiment results suggest that the proposed evaluation approach can provide more comprehensive insight into the strengths and weakness of CQA models.

**Questions For The Authors:**

Question A: Following reject reason 1. Could you discuss potential issues related to the proposed Interview Evaluation, such as bias, overfitting, or challenges in understanding and evaluating conversational question answering models?

**Reasons To Accept:**

1. The paper is commendable for its clear and fluent writing style. The authors effectively communicate complex ideas, which significantly aids understanding and the dissemination of their proposed method and findings.

2. The paper explores LLMs' abilities to conduct CQA evaluations, eliminating the need for expensive data collection of new datasets. The proposed idea is novel while practical and easy to implement on evaluating different CQA models. This represents a step forward in making CQA evaluations more accessible and cost-effective.

**Reasons To Reject:**

1. The authors propose using large language models to conduct CQA evaluations as an alternative to costly data collection. However, while LLMs are powerful, they can introduce biases present in their training data into their evaluations. Similarly, there could be issues of overfitting, where the model is excessively tailored to its training data and may not generalize well to new, unseen data. Furthermore, the ability of these models to understand and evaluate the aspects of conversational dynamics remains questionable. This lack of a discussion about these potential challenges undermines the paper's argument for the efficacy of using large language models for CQA evaluation.

2. In section 5, Adaptability and Flexibility for Interview Evaluation: The author rely solely on 'exact match' of  text to evaluate the diversity, which overlooks other factors of diversity. While this can show some level of diversity, it does not accurately or comprehensively reflect the actual depth of diversity and adaptability that the ChatGPT might be offering. Aspects such as question types, semantic meanings, context relevancy, and many other factors should be considered when evaluating diversity. Also, more details on how diversity correlates with model prediction are needed.

**Reproducibility:**

3: Could reproduce the results with some difficulty. The settings of parameters are underspecified or subjectively determined; the training/evaluation data are not widely available.

**Reviewer Confidence:**

3: Pretty sure, but there's a chance I missed something. Although I have a good feel for this area in general, I did not carefully check the paper's details, e.g., the math, experimental design, or novelty.

---

> ### Author Rebuttal · Authors · 2023-08-28
>
> We have thoroughly considered each of your suggestions and comments, and we now provide our detailed responses to address each point below:
>
>
>
> ----------Response to Comments----------
>
>
> **Comment #1:** The authors propose using large language models to conduct CQA evaluations as an alternative to costly data collection. However, while LLMs are powerful, they can introduce biases present in their training data into their evaluations. Similarly, there could be issues of overfitting, where the model is excessively tailored to its training data and may not generalize well to new, unseen data. Furthermore, the ability of these models to understand and evaluate the aspects of conversational dynamics remains questionable. This lack of a discussion about these potential challenges undermines the paper's argument for the efficacy of using large language models for CQA evaluation.
>
>
> **Response to #1:** We divide your comment into three aspects: Bias, Overfitting and The ability of these models to understand and evaluate, and reply separately.
>
>
> * **Bias**
>
> The issue you raised is highly relevant, as models like ChatGPT can exhibit certain biases. We believe that ChatGPT, trained on large-scale corpora with extensive knowledge and fine-tuned through RLHF, aligns with human evaluations. Hence, we employ a Q agent consisting of ChatGPT to replace human involvement in the evaluation process. Even in human evaluations, the generation of new questions may vary among individuals and different models, potentially introducing human biases. Our experiments demonstrate the alignment between our evaluation approach and human assessments (lines 552-566).
>
>
> * **Overfitting**
>
>
> We think that ChatGPT is sensitive to prompts. In our approach, we include instructions such as "give the interviewee some new hints" in the prompts to mitigate overfitting and encourage the model to generate novel outputs.
>
>
> * **The ability of these models to understand and evaluate**
>
>
> We propose two metrics, ACR and QPR, which effectively evaluate a model's understanding in dynamic dialogues. QPR measures the average number of interactions required for the model to correctly answer a question, with lower values indicating faster and more accurate responses. Additionally, in interview evaluations, the Q agent generates new prompting questions to guide the model towards providing correct answers. ACR captures this ability by assessing the model's capacity to refine its answers based on new feedback, mimicking the way humans refine their responses in dialogues (lines 330-340).
>
>
> **Comment #2:** In section 5, Adaptability and Flexibility for Interview Evaluation: The author rely solely on 'exact match' of text to evaluate the diversity, which overlooks other factors of diversity. While this can show some level of diversity, it does not accurately or comprehensively reflect the actual depth of diversity and adaptability that the ChatGPT might be offering. Aspects such as question types, semantic meanings, context relevancy, and many other factors should be considered when evaluating diversity. Also, more details on how diversity correlates with model prediction are needed.
>
>
> **Response to #2:** We fully agree with your perspective. In our analysis, diversity is considered as a subordinate feature, used to assess whether the G agent in interview evaluations is adaptable to different tested models, and doesn’t determine the quality of generated questions. We acknowledge that EM may impose overly strict limitations on diversity assessment. In future work, we plan to incorporate semantic evaluations such as BERTScore to address this limitation. We are actively working on this aspect.
>
>
>
>
>
> ----------Response to Questions----------
>
>
>
> **Question #1:** Following reject reason 1. Could you discuss potential issues related to the proposed Interview Evaluation, such as bias, overfitting, or challenges in understanding and evaluating conversational question answering models?
>
>
> **Response to #1:** The potential issues related to the proposed Interview Evaluation is worth discussing.
>
>
> Regarding bias, we believe that, by RLHF, ChatGPT works with a higher similarity to the preference of human, including the behavior of evaluating test results. On the other side, unfortunately, ChatGPT may produce biases similar to those present in the evaluations of human. We argue that their influence might not be severe.
>
>
> Regarding overfitting, it might be a crucial consideration in our evaluation methodology. it can restrict the selection of the question set in the Q agent. If the selected question set includes knowledge specific to the years 2023, our Q agent may encounter challenges in generating appropriate questions.
>
> Regarding the last challenge, we propose ACR and QPR as metrics. These metrics effectively evaluate the model's ability to refine its answers based on new feedback (ACR), and measure the average number of interactions required for accurate responses (QPR).
>
>
>
>
>
> ----------Response to Typos----------
>
> I'm glad that you found the suggestions helpful for improving readability. We will make every effort to rectify and improve the quality of future work. Thank you for bringing this to our attention.

---

### Official Review · Reviewer_uuh1 · 2023-08-04

**Soundness:** 3

**Excitement:**

3: Ambivalent: It has merits (e.g., it reports state-of-the-art results, the idea is nice), but there are key weaknesses (e.g., it describes incremental work), and it can significantly benefit from another round of revision. However, I won't object to accepting it if my co-reviewers champion it.

**Paper Topic And Main Contributions:**

In this paper, the authors propose an evaluation architecture for conversational question answering. Existing evaluation methods use a history of manually generated correct answers, which does not match the history predicted by the model. In contrast, the proposed method uses a question generator that generates questions according to the model's predictions without losing naturalness. The proposed method mimics an interview, where the interviewer asks questions to the model. If the model fails to answer, the interviewer generates other questions based on the previous exchanges. The authors also propose a rating scale according to this method.

**Reasons To Accept:**

- The proposed method considers the actual conversational situation that the answerer's answer may lead to asking supplementary questions.
- They propose several evaluation metrics that allow the evaluation of various aspects of the model.

**Reasons To Reject:**

- The model's performance depends on the performance of the generator G, but the validity of the questions generated by G has yet to be examined.
- They have analyzed the model using the three metrics they proposed, but the validity of these metrics themselves has not been adequately examined.

**Reproducibility:**

2: Would be hard pressed to reproduce the results. The contribution depends on data that are simply not available outside the author's institution or consortium; not enough details are provided.

**Reviewer Confidence:**

2: Willing to defend my evaluation, but it is fairly likely that I missed some details, didn't understand some central points, or can't be sure about the novelty of the work.

**Typos Grammar Style And Presentation Improvements:**

- What is stated in lines 554-558 of the text is inconsistent with Table 3.

---

> ### Author Rebuttal · Authors · 2023-08-28
>
> We apologize for any reading difficulties caused by the typos in the paper. We take your feedback seriously, and our response is as follows:
>
>
>
> ----------Response to Comments----------
>
>
> **Comment #1:** The model's performance depends on the performance of the generator G, but the validity of the questions generated by G has yet to be examined.
>
>
> **Response to #1:** The issue you raised is indeed significant, as ensuring the validity of generated questions poses a challenge.
>
>
> First, we compare the metrics obtained after applying method G with human evaluations, demonstrating alignment between our proposed interview evaluation using real dialogue history and human evaluation. In lines 553-566, we discuss the alignment with human evaluation, validating our proposed evaluation metrics.
>
>
> Second, in our paper, the generator G is ChatGPT. Fang et al (2023) [1] stated that "we can conclude that ChatGPT generates grammatically corrected sentences that are highly fluent". Therefore, we hold the belief that the generated question exhibits a high level of grammatical quality. Furthermore, the configuration of G used for generating questions was kept fixed across different models being evaluated. Therefore, any impact resulting from generating grammar errors or lack of fluency in questions can be considered as a systematic error that does not alter the ranking of the evaluated models.
>
>
> Third, our approach involves manual analysis of the questions generated by the generator G. We conducted approximately 20 iterations of prompt tuning, to determine the final prompt. We show several prompts in Table 1 (shown in the Appendix at the end of response), including those with improper instruction settings and lacking information. We check the generated questions in 50 dialogues of each prompt, and compare the effect of prompts by manually checking the fluency, plausibility and relevance of generated questions.
>
>
> Finally, allow me to further elaborate on the necessity of introducing G. The automatic evaluation of the dialogue ability of the CQA model has always faced the problem of static evaluation [2,3]. We introduced interactivity in the automatic evaluation of CQA models by introducing G, and designed three indicators based on this feature.
>
>
> In future iterations or expanded versions of the paper, we intend to conduct a more extensive quality evaluation, incorporating methodologies to ensure generator quality.
>
>
> **Comment #2:** They have analyzed the model using the three metrics they proposed, but the validity of these metrics themselves has not been adequately examined.
>
>
> **Response #2:** We cite the evaluation accuracy (Acc) scores for the BERT, GraphFlow, HAM, and ExCorD models from "Ditch the Gold Standard: Re-evaluating Conversational Question Answering" and calculate Spearman and Pearson correlation scores between these metrics and our QPR and PFR metrics. This demonstrates the alignment between our metrics and human evaluations.
>
>
> Our three metrics are tailored to the interactive nature of interview evaluations, where the Q agent plays a role in guiding the model to provide correct answers. The three metrics are characterized as below:
>
> QPR can be used to measure the average number of interactions required for the model to correctly answer a question. A lower QPR indicates the model's ability to answer questions more quickly.
>
>
> PFR can be used to evaluate the model's capability to successfully respond to questions during interactions. A higher PFR indicates more frequent failures in answering questions, while a lower PFR indicates fewer failures.
>
>
> ACR is a dynamic evaluation method. When a model refuses to answer a question, the Q agent generates new probing questions to guide the model towards providing the correct answer. We measure this ability using ACR, which reflects the model's capacity to refine its answers based on new feedback, similar to how humans refine their responses in dialogues (lines 330-340).
>
>
> ----------Response to Typos----------
>
>
> **Typos #1:** What is stated in lines 554-558 of the text is inconsistent with Table 3.
>
>
> **Response to #1:** We apologize for any confusion caused by the error in the description of Table 3. The ♣ symbol corresponds to the metrics for interview evaluation based on generated history, while the ♢ symbol corresponds to the metrics for interview evaluation based on golden history. The description in lines 557-558 was incorrect. In terms of QPR, both the golden-history interview evaluation and human evaluation exhibit strong agreement, with a Spearman correlation coefficient of -1.0 and a Pearson correlation coefficient of -0.91. Apologies for any confusion caused by a typo in the formula, where the denominator should be $N_s$ in line 322.
>
>
>
> ----------Reference----------
>
>
> [1] Fang, Tao and Yang, Shu and Lan, Kaixin and Wong, Derek F and Hu, Jinpeng and Chao, Lidia S and Zhang, Yue. Is chatgpt a highly fluent grammatical error correction system? a comprehensive evaluation. In arxiv, 2023.
>
> [2] Li, Huihan and Gao, Tianyu and Goenka, Manan and Chen, Danqi. Ditch the gold standard: Re-evaluating conversational question answering. In ACL, 2022.
>
> [3] Deng, Yang and Lei, Wenqiang and Huang, Minlie and Chua, Tat-Seng. Goal Awareness for Conversational AI: Proactivity, Non-collaborativity, and Beyond. In ACL, 2023 tutorials.1.
>
>
> ----------Appendix----------
>
>
> **Tabel 1 Comparison of Generated Questions for Each Prompt**
>
> We provide three examples of question generation for each prompt, with the "origin question" column presenting three questions from the question set, and the "generated question" column showcasing three questions generated using the corresponding prompt.
>
>    | Prompt |Origin questions| Generated questions| Performance |
>    | ------ | -------|----|--------- |
>    | You play a interviewer and someone else plays a interviewee, you ask a question, and if you judge the interviewee to answer correctly,. Otherwise you need to prompt the interviewee for the wrong answer and give the interviewee some new hints based on the correct answer without revealing the answer.this is history conversatio{history}\nThen when you ask {cur_question}, the interviewee answer {cur_answer},however the gold answer is {gold_answer}. | 1.What was In This Light? 2.Did Shula lose to any coaches?3.What happened next in their business? | 1.Interviewer: Welcome to the interview! ...Interviewee: ... . Music Technology at Staffordshire University. 2.Interviewer: Can you tell me about Don Shula's coaching career?Interviewee: ... .  3.Interviewer: Let's dive into our history conversation...Interviewee: ... | negative    |
> | [               {'role':'system','content':"""This is a turing test. You play a interviewer and someone else plays a interviewee, you ask a question, and if you judge the interviewee to answer correctly,. Otherwise you need to prompt the interviewee for the wrong answer and give the interviewee some new hints based on the correct answer without revealing the answer."""}, {'role':'user','content':f"""this is history conversation{history}""" },{'role':'system','content':f"""Then when you ask {cur_question}, the interviewee answer {cur_answer},however the gold answer is {gold answer}."""}] | 1.What was In This Light? 2.Did Shula lose to any coaches? 3.What happened next in their business? | 1.Thank you .... Can you tell me about the significance of the Magna Carta in English history? 2.That's close, but not quite the answer I was looking for Don Shula did have Super Bowl losses, but I was actually referring to his regular season record against other coaches... . 3.Thank you for that response. While it is true that Orson Welles worked extensively in radio as an actor, ...? | negative |
> | [{'role':'system','content':"""This is a turing test. You play a interviewer and someone else plays a interviewee, you ask a question, and if you judge the interviewee to answer correctly, you keep asking questions. Otherwise you need to prompt the interviewee for the wrong answer and give the interviewee some new hints based on the correct answer without revealing the answer. The question is a single-hop question and end the conversation, and should not be redundant."""}, {'role':'user','content':f"""this is history conversation {history}"""},              {'role':'system','content':f"""Then when you ask '{cur_question}', the interviewee answer '{cur_answer}',. What question do you ask to prompt interviewee? The question is a single-hop and short!"""}] | 1.What was In This Light? 2.Did Shula lose to any coaches? 3.What happened next in their business? | 1.Can you provide more details about the band's formation and how they came together while studying Music Technology at Staffordshire University? 2.Can you provide more information about Don Shula's record against Vince Lombardi? 3.Can you provide an example of a radio production that Orson Welles worked on during this time? | negative    |
> | [{'role':'system','content':"""This is a turing test. You play a interviewer and someone else plays a interviewee, you ask a question, and if you judge the interviewee to answer correctly, you keep asking questions. Otherwise you need to prompt the interviewee for the wrong answer and give the interviewee some new hints based on the correct answer without revealing the answer. The question is a single-hop question and end the conversation, and should not be redundant."""}, {'role':'user','content':f"""this is history conversation {history}"""},{'role':'system','content':f"""Then when you ask '{cur_question}', the interviewee answer '{cur_answer}', however the gold answer is '{gold answer}'. What question do you ask to prompt interviewee? The question is a single-hop and short!"""} ] | 1.What was In This Light? 2.Did Shula lose to any coaches? 3.What happened next in their business? | 1.Can you tell me the name of the latest album released by the band?  2.Can you name any other coaches that Don Shula had losing records against? 3.What famous radio adaptation brought Welles instant fame? | positive |
>
> Thank you again for your insightful feedback, which undoubtedly contributes to enhancing the quality and enriching the final version of the paper.

---

### Official Review · Reviewer_5zAa · 2023-08-07

**Soundness:** 3

**Excitement:**

3: Ambivalent: It has merits (e.g., it reports state-of-the-art results, the idea is nice), but there are key weaknesses (e.g., it describes incremental work), and it can significantly benefit from another round of revision. However, I won't object to accepting it if my co-reviewers champion it.

**Paper Topic And Main Contributions:**

The paper proposes a novel evaluation method for conversational question answering models (CQA) by generating related subsequent questions for wrong answers from a CQA model. This approach aims to provide more opportunities for the model to answer correctly by asking detailed and generated questions, simulating an interview-like setting. The authors introduce three metrics to quantify this evaluation: questions per round (QPR, higher is better), persistent failure rate (PFR, lower is better), and answer conversion rate (ACR, higher is better). The proposed evaluation methods are tested on two CQA datasets – QuAC and CoQA – and their performance is compared to human evaluation accuracy. The results demonstrate that the suggested methods partly align with human evaluation results, and the paper also investigates various aspects of the evaluation approach.

**Questions For The Authors:**

-	How can "reference and omission" be measured using an evaluation method not covered well in Li et al. (2022)? (lines 147 – 150)
-	Does the generator in the Q agent implicitly know about the questions? For example, during an interview, an interviewer might know the answer to a question, leading them to generate more appropriate subsequent questions. Similarly, does ChatGPT know the answers to every question?
-	What is the configurable threshold for overlap between the model prediction and the golden answer? Does this threshold significantly affect the performance of the evaluation metrics? (lines 259 – 253)


**Reasons To Accept:**

One of the significant contributions of this paper is the interesting and straightforward method for evaluating conversational question answering models. Evaluation methods for natural language generation (NLG) tasks are challenging yet crucial, and the suggested idea can be extended to other NLG tasks, such as dialogue systems, due to its simplicity.

**Reasons To Reject:**

The main concern revolves around the measurement of the appropriateness of the proposed evaluation methods. It is challenging to determine whether QPR, PFR, and ACR align with human evaluation accuracy as shown in Table 1. Furthermore, it is unclear how the human evaluation results in Table 3 were conducted. The authors should provide more details on the human evaluation process and ensure that it reflects the perspective of the suggested evaluation methods. For instance, if evaluating fluency, human evaluators should be asked to assess grammar errors and fluency explicitly. So they can set up the fine-grain questions for showing the comprehensive insights.

**Reproducibility:**

3: Could reproduce the results with some difficulty. The settings of parameters are underspecified or subjectively determined; the training/evaluation data are not widely available.

**Reviewer Confidence:**

3: Pretty sure, but there's a chance I missed something. Although I have a good feel for this area in general, I did not carefully check the paper's details, e.g., the math, experimental design, or novelty.

---

> ### Author Rebuttal · Authors · 2023-08-28
>
> Your feedback is highly valued and has undoubtedly played a crucial role in enhancing the quality of our work. We have carefully considered each of your suggestions and comments, and we provide our responses to address your concerns below:
>
>
>
> ----------Response to Comments----------
>
>
> **Comment #1:** The main concern revolves around the measurement of the appropriateness of the proposed evaluation methods. It is challenging to determine whether QPR, PFR, and ACR align with human evaluation accuracy as shown in Table 1. Furthermore, it is unclear how the human evaluation results in Table 3 were conducted. The authors should provide more details on the human evaluation process and ensure that it reflects the perspective of the suggested evaluation methods. For instance, if evaluating fluency, human evaluators should be asked to assess grammar errors and fluency explicitly. So they can set up the fine-grain questions for showing the comprehensive insights.
>
>
> **Response to #1:** We divide your comment into two aspects: Alignment with Human Evaluation, Details on the Human Evaluation Process, and reply separately.
>
>
> * **Alignment with Human Evaluation**
>
>
> In our paper (lines 553-566), we have conducted statistical analyses using Pearson and Spearman correlation coefficients to establish the alignment between the interview evaluation metrics and human evaluation. We deeply apologize for any inconvenience caused by the typos that have appeared in the original text, and we will address and correct all the issues in the expanded version. In a word, we observed a strong negative correlation (Spearman's $\rho$ = -1.0,  Pearson's $r$ = -0.91) in the QPR metrics of the golden-history interview evaluation. The closer the correlation coefficients is to 1 or -1, the more aligned it is, while the closer it is to 0, the less aligned it is. These results demonstrate the alignment between our proposed metrics and human evaluations, supporting the reliability of our evaluation approach.
>
>
> * **Details on the Human Evaluation Process**
>
>
> We apologize for the oversight in excluding the details of the human evaluation process in our initial submission. The human evaluation metrics for the four CQA models were referenced from Table 2 of the paper Li et al (ACL, 2022) [1]. The human evaluation process described in the referenced paper involved three main steps conducted on the Amazon Mechanical Turk platform.
>
>
> Firstly, the annotators were provided with questions without access to the passage. Secondly, after the conversation end, the annotators were shown with the passage and asked to verify the correctness of the model predictions. Lastly, two additional annotators were involved to validate the annotations.
>
>
> The referenced paper has been widely used in CQA evaluation, including being referenced in the ACL 2023 tutorial [2] on "Open Challenges for Conversational Agents' Awareness and Beyond: Evaluation for Conversational Agent's Awareness."
>
>
> We believe that the key advantage of human evaluation lies in introducing the interaction between humans and machines during the evaluation process. This insight inspires us to incorporate this interactivity in our automatic evaluation as well.
>
>
> Thank you for acknowledging the specific instances provided. Grammar errors and fluency are indeed crucial dimensions for evaluation. Due to the answers extracted from well-written articles, such as Wikipedia or news sources, we assume that the answers are grammatically correct and fluent. Therefore, the explicit evaluation of grammar errors and fluency is not within the scope of our evaluation.
>
>
>
> ----------Response to Questions----------
>
>
> **Question #1:** How can "reference and omission" be measured using an evaluation method not covered well in Li et al. (2022)? (lines 147–150)
>
>
> **Response to #1:** We apologize for any confusion. Li et al. (2022) introduced three evaluation methods: human evaluation and two automated methods simulating human evaluation. While human evaluation naturally captures linguistic phenomena like reference and omission in human-machine dialogues, the two automated methods deviate from assessing CQA models based on "reference and omission" in conversational questions. In our interview evaluation, we manually analyzed 20 conversations, comprising 141 questions. Among them, 39 questions exhibited omission, and 93 questions involved reference phenomena. This observation confirms the inclusion of reference and omission properties in the our interview evaluatetion.
>
>
> **Question #2:** Does the generator in the Q agent implicitly know about the questions? For example, during an interview, an interviewer might know the answer to a question, leading them to generate more appropriate subsequent questions. Similarly, does ChatGPT know the answers to every question?
>
>
> **Response to #2:** We explicitly provided ChatGPT with the answers to every question.
>
>
> **Question #3:** What is the configurable threshold for overlap between the model prediction and the golden answer? Does this threshold significantly affect the performance of the evaluation metrics? (lines 259–253)
>
>
> **Response to #3:** The threshold represents the level of lexical overlap between predictions and gold answers, aligned with downstream tasks. If a downstream task demands high similarity between predicted and gold answers, the threshold can be increased. In our proposed interview evaluation, if the overlap between prediction and gold answer exceeds the configurable threshold (0.5), the current question is considered successfully answered, and the question generation process is terminated. Therefore, the configurable threshold impacts the QPR and FPR metrics. Configurable parameters are common in evaluations, such as binary classification F-score.
>
>
>
> ----------Reference----------
>
>
> [1]Li, Huihan and Gao, Tianyu and Goenka, Manan and Chen, Danqi. Ditch the gold standard: Re-evaluating conversational question answering. In ACL, 2022.
>
>
> [2]Deng, Yang and Lei, Wenqiang and Huang, Minlie and Chua, Tat-Seng. Goal Awareness for Conversational AI: Proactivity, Non-collaborativity, and Beyond. In ACL, 2023 tutorials.1.
>
>
> We sincerely appreciate the constructive critique provided by the reviewers, which has undeniably enriched the final version of the paper. Thank you for your time and consideration.

---

### Meta-Review · Area_Chair_W5Cx · 2023-09-15

**Recommendation:** 4

**Metareview:**

This paper advocates a novel method for evaluating QA models, especially those dialog-oriented: the authors propose an architecture, relying on Chat-GPT, to conduct an "interview" with a QA system. During the interview, Chat-GPT augments a pre-recorded sequence of questions (interview script) with generated hints and paraphrased Qs in order to nudge the system to arrive at a correct answer. This has several advantages: (1) it can reduce the cost of expensive human-based evaluation and (2) it can considerably improve our understanding of QA performance according to different criteria.

All the reviewers agree that the idea is novel and definitely interesting. Moreover, it is very impactful and this research would be immediately beneficial for the NLG community.

The paper is written very well and provides multiple insights on the proposed metrics, highlighting their meaning, interpreting the results on two datasets and discussing alignment with human judgement.

Two major issues have been raised by the reviewers. First, all the reviewers have given middle-low reproducibility scores and raised specific technical issues. Generally speaking, the authors provide a lot of technical details in the appendix -- yet, some questions remain.  I believe it is absolutely crucial for evaluation papers to arrive at maximum reproducibility, otherwise effective adoption of proposed metrics by the community is impossible. The authors, however, have very thoroughly clarified these issues in the response. One particular issue: the algorithm depends on the specific prompt template. The template is not presented in the paper -- but was revealed by the authors in the discussion. The omission of the template from the final version would considerably undermine reproducibility. (Figure 7 in the Appendix provides a very good idea of the template, yet it might be insufficient to reproduce the results.)

Second, reviewers uuh1 and dejG argue that the approach relies crucially on the properties of the interviewer (Chat-GPT). This might introduce multiple effects, from non-valid questions to biases etc. This is a more general problem that requires an in-depth discussion -- and the authors have provided some insights in the rebuttal, to be incorporated into the paper. I believe, however, that this issue is not a weakness per se, but rather a starting point for important further research.

---

### Decision · Program_Chairs · 2023-10-07

**Decision:**

Accept-Main

**Comment:**

This paper advocates a novel method for evaluating QA models, especially those dialog-oriented: the authors propose an architecture, relying on Chat-GPT, to conduct an "interview" with a QA system. During the interview, Chat-GPT augments a pre-recorded sequence of questions (interview script) with generated hints and paraphrased Qs in order to nudge the system to arrive at a correct answer. This has several advantages: (1) it can reduce the cost of expensive human-based evaluation and (2) it can considerably improve our understanding of QA performance according to different criteria.

All the reviewers agree that the idea is novel and definitely interesting. Moreover, it is very impactful and this research would be immediately beneficial for the NLG community.

The paper is written very well and provides multiple insights on the proposed metrics, highlighting their meaning, interpreting the results on two datasets and discussing alignment with human judgement.

Two major issues have been raised by the reviewers. First, all the reviewers have given middle-low reproducibility scores and raised specific technical issues. Generally speaking, the authors provide a lot of technical details in the appendix -- yet, some questions remain.  I believe it is absolutely crucial for evaluation papers to arrive at maximum reproducibility, otherwise effective adoption of proposed metrics by the community is impossible. The authors, however, have very thoroughly clarified these issues in the response. One particular issue: the algorithm depends on the specific prompt template. The template is not presented in the paper -- but was revealed by the authors in the discussion. The omission of the template from the final version would considerably undermine reproducibility. (Figure 7 in the Appendix provides a very good idea of the template, yet it might be insufficient to reproduce the results.)

Second, reviewers uuh1 and dejG argue that the approach relies crucially on the properties of the interviewer (Chat-GPT). This might introduce multiple effects, from non-valid questions to biases etc. This is a more general problem that requires an in-depth discussion -- and the authors have provided some insights in the rebuttal, to be incorporated into the paper. I believe, however, that this issue is not a weakness per se, but rather a starting point for important further research.